# Synthesis of propenone-linked covalent organic frameworks via Claisen-Schmidt reaction for photocatalytic removal of uranium

Cheng-Peng Niu[1,3], Cheng-Rong Zhang[1,3], Xin Liu[1], Ru-Ping Liang ●[1]✉ & Jian-Ding Qiu ●[1,2]✉

The type of reactions and the availability of monomers for the synthesis of $sp^2$-c linked covalent organic frameworks (COFs) are considerably limited by the irreversibility of the C=C bond. Herein, inspired by the Claisen-Schmidt condensation reaction, two propenone-linked (C=C−C=O) COFs (named Py-DAB and PyN-DAB) are developed based on the base-catalyzed nucleophilic addition reaction of ketone-activated $\alpha$-H with aromatic aldehydes. The introduction of propenone structure endows COFs with high crystallinity, excellent physicochemical stability, and intriguing optoelectronic properties. Benefitting from the rational design on the COFs skeleton, Py-DAB and PyN-DAB are applied to the extraction of radionuclide uranium. In particular, PyN-DAB shows excellent removal rates (>98%) in four uranium mine wastewater samples. We highlight that such a general strategy can provide a valuable avenue toward various functional porous crystalline materials.

Covalent organic frameworks (COFs) represent a class of advanced porous crystalline materials, which specifically integrate various functional units into highly ordered periodic arrays and feature highly customizable structure and functions[1,2]. In recent years, COFs based on irreversible C=C bond connection have attracted extensive attention for their excellent stability and π-electronic communication characteristics[3], which overcome the disadvantages of borate ester or dynamic imine linked COFs in this respect[4]. Therefore, they have considerable application potential in energy[5], environment and health fields[6,7].

However, the relatively poor reversibility of C=C bonds makes the synthesis of $sp^2$ carbon-conjugated COFs challenging, limiting viable strategies to only a few reactions, including the Aldol reaction[8], the Knoevenagel reaction[9], and the Horner-Wadsworth-Emmons polycondensation[10]. However, the monomers used in these reactions must contain numerous electron-withdrawing groups[9,11], which

requires cumbersome process and even unavoidably introduces toxic raw materials[12]. The design and synthesis of $sp^2$ carbon-conjugated COFs will continue to be at the forefront of COFs research as their unparalleled potential for combination in crystallinity, porosity, stability, modularity, and functionality.

Claisen-Schmidt condensation has been extensively used in organic chemistry to introduce C=C bonds[13,14]. Compared to the classical reaction of olefin bond formation described above, the essence of the Claisen-Schmidt reaction is that the $\alpha$-H activated by the ketone structure, which condensates with the aromatic aldehyde under the catalysis of base to form a stable C=C bond. Thus, the monomers used in Claisen-Schmidt reaction are free of cyano and more environmentally friendly. In addition, such the compound formed by Claisen-Schmidt condensation with a propenone (C=C−C=O) structure contains a large π-bond, which makes such materials exhibit excellent photoelectronic performance. Furthermore, introducing ketone

[1]School of Chemistry and Chemical Engineering, Nanchang University, Nanchang 330031, China. [2]State Key Laboratory of Nuclear Resources and Environment, East China University of Technology, Nanchang 330013, China. [3]These authors contributed equally: Cheng-Peng Niu, Cheng-Rong Zhang. ✉e-mail: rpliang@ncu.edu.cn; jdqiu@ncu.edu.cn

structure can enhance the light absorption[15], and the lone pair electron resonance effect of oxygen atom can improve the stability of materials[16], which will further improve designability and functionality of the interrelated materials.

Herein, we report a strategy for synthesizing unsubstituted propenone-linked COFs via the base-catalyzed Claisen-Schmidt reaction between aryl acetyl and aryl aldehydes. Two $sp^2$-$c$ conjugate COFs (Py-DAB and PyN-DAB) with high crystallinity were successfully synthesized by combining 1,4-diacetophenone (DAB) with 1,3,6,8-tetra-kis(4-formylphenyl)pyrene (TFPPy) and 5,5',5'',5'''-(pyrene-1,3,6,8-tetrayl)tetrapicolinaldehyde (TFPPyN), respectively. Py-DAB and PyN-DAB exhibited excellent physicochemical stability and photocatalytic performance due to the introduction of the propenone structure. In particular, benefitting from the rational design on the COFs skeleton, PyN-DAB exhibited a higher radioactive uranium adsorption capacity under visible light through photocatalytic reduction mechanism, and showed excellent removal rates (>98%) in four uranium mine wastewater samples, which overcome the disadvantages of the complex post-modification process of amidoxime-based COFs and the poor selectivity produced by vanadium. We highlighted that such a strategy will enrich the types of $sp^2$-$c$ COFs and provide a valuable avenue toward various functional porous crystal materials.

## Results

### Synthesis and characterization of COFs

Initially, the model compound was successfully prepared by the classical method to verify the feasibility of the reaction (Supplementary Fig. 1). The proposed mechanism is that the α-H on acetophenone can be activated by the ketone structure, deprotonated under the catalysis of alkali, and undergoes a nucleophilic addition reaction with aromatic aldehydes to form a stable C=C bond (Fig. 1a). Subsequently, we have conducted extensive screening of the synthesis conditions to explored the reaction between 1,3,6,8-tetrakis(4-formylphenyl)pyrene (TFPPy) and 1,4-diacetophenone (DAB) (Fig. 1b, Supplementary Table 1, Supplementary Figs. 2, 3). Finally, the crystalline condensation product (Py-DAB) was synthesized under the optimal reaction conditions (more details in the Supporting Information). To further validate the versatility of Claisen-Schmidt reaction for the synthesis of COFs, we chose 5,5',5'',5'''-(pyrene-1,3,6,8-tetrayl)tetrapicolinaldehyde (TFPPyN) to react with DAB to produce PyN-DAB (Fig. 1b, Supplementary Table 2, Supplementary Figs. 4, 5), and the results were worth noting that Py-DAB and PyN-DAB were synthesized under mild conditions in high yields.

To evaluate the crystallinity of Py-DAB and PyN-DAB, the powder X-ray diffractometry (PXRD) measurement was examined. As shown in

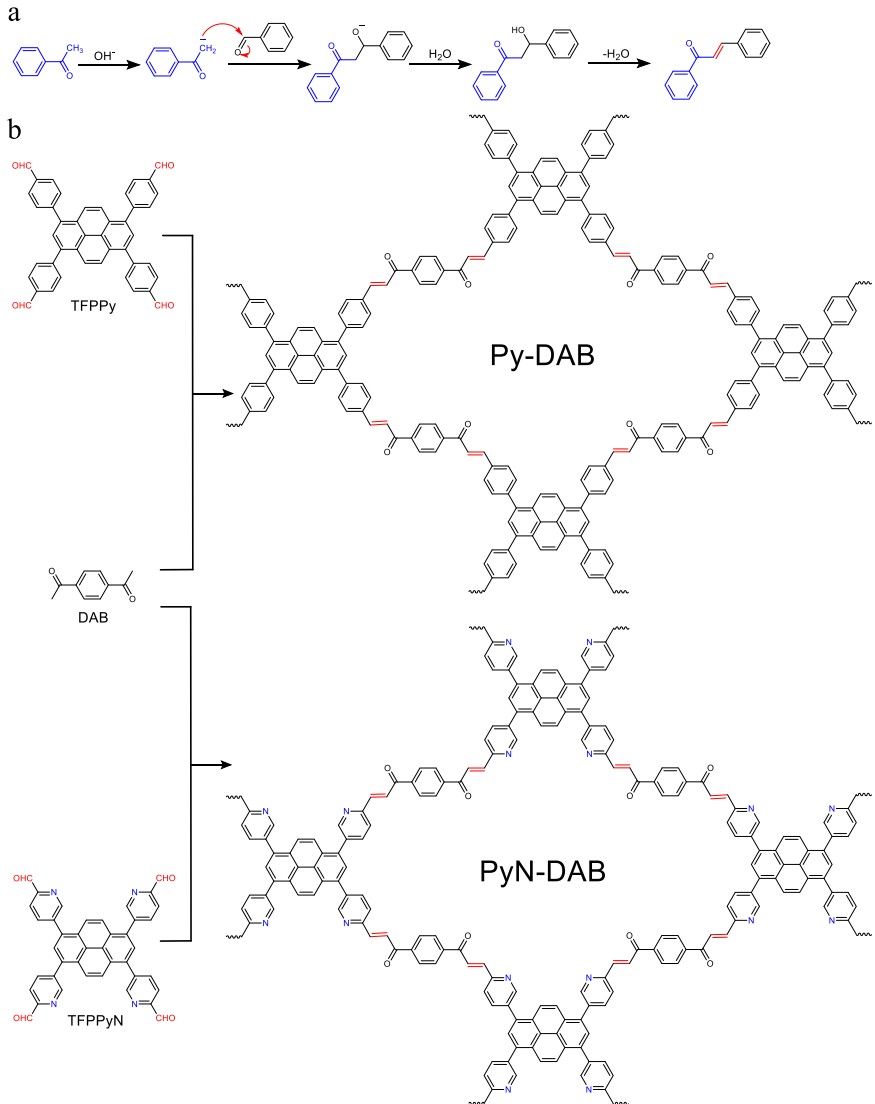

**Fig. 1 | Schematic of synthetic propenone-linked COFs. a** The mechanism of C=C bonds formation by the Claisen-Schmidt reaction. **b** Synthesis of propenone-linked COFs.

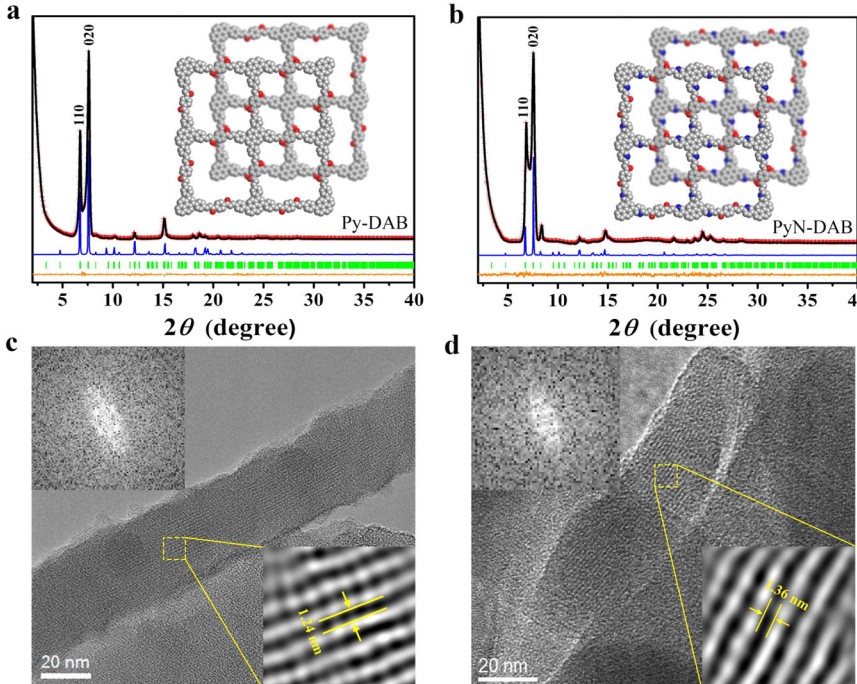

**Fig. 2 | Structural representations of propenone-linked COFs.** PXRD patterns of **a** Py-DAB and **b** PyN-DAB with the experimental (red cross) and Pawley refined (black line) profiles, the refinement difference (orange line), the eclipsed AB stacking model (blue line) and the Bragg position (green bar). (Insets: eclipsed AB stacking model, C: gray, O: red, N: blue). HR-TEM images of **c** Py-DAB and **d** PyN-DAB. Upper left: FFT pattern. Low right: enlarged view of a selected area.

Fig. 2a, b, both Py-DAB and PyN-DAB showed strong diffraction peaks at low angle, which confirmed their long-range ordered framework structure. To verify the stacking mode of COFs, AA and AB stacking models were simulated by Material Studio (Fig. 2a, b, Supplementary Figs. 6, 7), and the density functional bonding (DFTB) method was used to quantitatively evaluate the interlayer interaction. The results showed that AB mode with higher stacking energy (523.83 kcal mol$^{-1}$) than AA mode (428.75 kcal mol$^{-1}$), proving the staggered AB stacking structure is much more stable[17]. This may be attributed to the strong dipole moment generated by the heteroatoms (oxygen atoms) in the skeleton to avoid these charged atoms directly located at the top of each other[4]. The Pawley refinement for these COFs yielded PXRD patterns showed reasonable agreement with the simulation of AB mode (Supplementary Tables 3 and 4), which was similar to previous reports[18,19]. Additionally, Py-DAB with negligible residuals Rwp = 5.58% and Rp = 2.70%, and PyN-DAB with negligible residuals Rwp = 3.84% and Rp = 1.27%, which were attributed to the existence of crystalline 2D network[20]. The morphologies of COFs were characterized by scanning electron microscopy (SEM) (Supplementary Figs. 8, 9) and transmission electron microscopy (TEM), and the TEM images confirmed that these two COFs have homogeneous rod-like shape (Supplementary Figs. 10, 11). The HR-TEM images showed clear lattice fringes, proving these two COFs have the high crystallinities. In addition, one-dimensional channels with uniform distance of 1.24 and 1.36 nm were clearly visualized, and the fast Fourier transform (FFT) of the original HR-TEM images produces a quadrilateral arrangement of four white diffraction spots (Fig. 2c, d), which further revealing the ordered internal structure.

The chemical structures of Py-DAB and PyN-DAB were confirmed by Fourier transform infrared (FTIR) spectroscopy. Clearly, the stretching vibration peak of Ar-CHO (ca. 1697 cm$^{-1}$) was greatly reduced[21]. The vibration peak of ketone structure (ca. 1672 cm$^{-1}$) had a certain degree of retention, and the vibration peak of *trans* -C=C- (ca. 970 cm$^{-1}$) newly appeared (Supplementary Figs. 12, 13)[22], indicating that aldehyde groups and methyl groups were highly condensed. In

addition, the energy distribution of the reactions in the different states was calculated to provide a basis for comparing their thermodynamically favorable reactions (Supplementary Figs. 14, 15). Comparing the *trans* and *cis* pathways in the case of alkali, the most important reaction is step 3, in which the C–C bond formation occurs and determines the preferred geometry and associated facial selectivity[23]. Clearly, this step is an exothermic reaction for the formation of the *trans* structure, whereas for the *cis* structure this step is a heat-absorbing reaction. This implies that the formation of *trans* C=C bonds under base catalysis is thermodynamically more favorable than the formation of *cis* C=C bonds. To further indicate the efficient condensation, the $^{13}$C cross-polarization magic-angle spinning (CP/MAS) NMR spectroscopy was tested. As shown in Fig. 3a, a well-resolved peak located at about 144 ppm belonged to the newly formed olefin groups, the signal at around 135 ppm was typically arising from olefin-linked phenyl carbon[24], and the peak located at around 190 ppm was ascribed to the carbon atoms in the ketone groups[25]. Similarly, PyN-DAB also exhibited distinct characteristic peaks (Fig. 3b). The above results proved that Py-DAB and PyN-DAB were successfully synthesized.

The surface area and porosity of Py-DAB and PyN-DAB were assessed by N$_2$ sorption analysis at 77 K. The test results showed that the Brunauer-Emmett-Teller (BET) surface areas were calculated to be 633.5 and 456.3 m$^2$ g$^{-1}$, and the pore size distributions centered at 1.31 and 1.33 nm, respectively (Fig. 3c, d). In the AB stacking mode, the pore size of Py-DAB and PyN-DAB are reduced due to the mismatch of the original pores, but it is relatively concentrated and matched well with the simulated calculation by density functional theory (about 1.36 nm).

To evaluate the physicochemical stability of Py-DAB and PyN-DAB, each material was exposed to different harsh conditions, including strong acidity (9 M HCl), strong basicity (9 M KOH), and intense radiation (200 K Gy) for 24 h. The test results showed that all Py-DAB and PyN-DAB exhibited less remarkable cost of crystallinity under the above conditions by PXRD analysis (Supplementary Figs. 16, 17). In addition, the thermal stability of Py-DAB and PyN-DAB was characterized by thermal gravimetric analysis (TGA). The results showed that

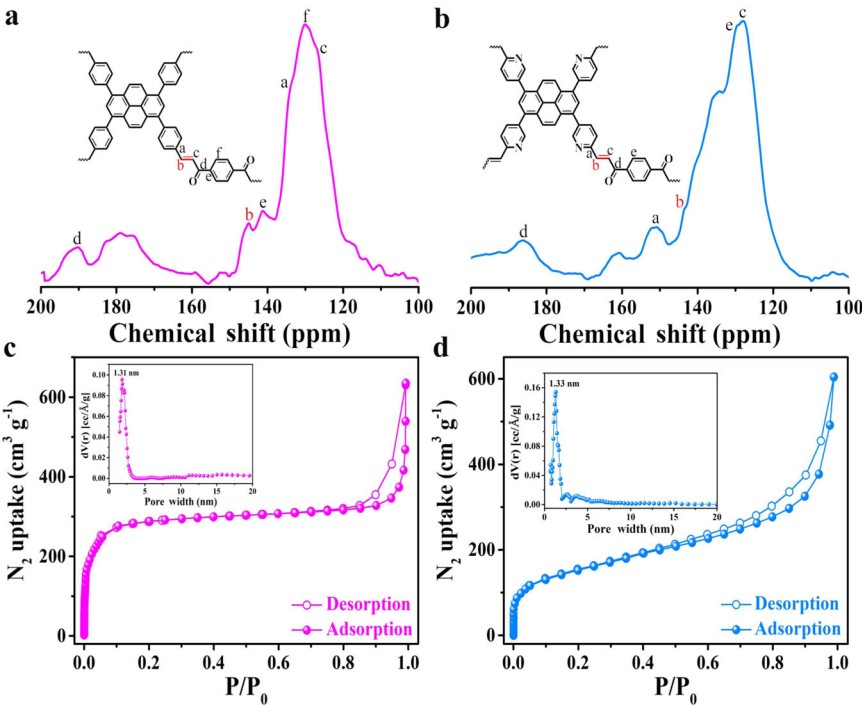

**Fig. 3 | Characterization of propenone-linked COFs.** The $^{13}$C cross-polarization magic-angle spinning (CP/MAS) NMR spectroscopy of **a** Py-DAB and **b** PyN-DAB. $N_2$ sorption isotherms of **c** Py-DAB and **d** PyN-DAB (Insets: pore size distributions).

Py-DAB and PyN-DAB exhibited excellent thermal stability up to 300 °C (Supplementary Figs. 18, 19).

## Characterization of propenone-linked COFs in photoelectric properties

Generally, the introduce of C=C bond and ketone structure gives the materials excellent chemical stability, π-electron communication property in the X-Y plane and a wide range of light collection[26]. Therefore, the semiconductor behavior and optoelectronic properties of the synthesized Py-DAB and PyN-DAB were studied systematically. First of all, their ultraviolet-visible diffuse reflection (UV-vis DRS) spectroscopy indicated that Py-DAB and PyN-DAB exhibited broad boundary absorption in the ultraviolet and visible regions, and their optical band gaps were determined to be 2.15 and 1.76 eV by the Kudelka-Munk transform reflection spectra (Fig. 4a), respectively. The absorption edge of PyN-DAB with wider absorption may be attributed to the introduce of strong electronegative nitrogen atoms, which forms the effect of *p-π* conjugation with *π* electrons in COFs[27]. The wide range of light absorption and narrow band gaps proved that Py-DAB and PyN-DAB can be used in practical photocatalytic applications. The fluorescence lifetimes of Py-DAB and PyN-DAB were measured to confirm the importance of propenone structure in promoting the separation and transportation of photogenerated carriers. The results of the average lifetimes were calculated to be 2.21 and 3.40 ns, respectively (Fig. 4b). The longer lifetime of Py-DAB and PyN-DAB demonstrated that the recombination of photogenerated carriers was further inhibited[28,29], which was related to its more conjugated *sp²-c* skeleton.

Mott-Schottky results showed that Py-DAB and PyN-DAB are all n-type semiconductors (Supplementary Figs. 20, 21), indicating that the main charge carriers are electrons[30]. The smaller semicircular domains in the electrochemical impedance spectroscopy (EIS) represent the lower electron transfer resistance (Ret) and higher charge transfer rate (Fig. 4c). To further elucidate the photoelectric properties of Py-DAB and PyN-DAB, the photoelectric response was tested. As shown in Fig. 4d, the results proved that Py-DAB and PyN-DAB had the

advantages of superior electro-optical activity, which suggesting they existed a much more efficient separation of interfacial charge. To further verify the excellent stability and photoelectric performance caused by the introduction of propenone structure, an imine-linked COF (Py-PD) was synthesized and tested as a comparison, and the results showed that it exhibited poor stability and photoelectric performance compared to Py-DAB and PyN-DAB (Supplementary Fig. 22).

In addition, the repeat unit of the Py-DAB and PyN-DAB was selected and conducted quantum chemistry calculations by density functional theory (DFT). The results show the bandgap narrowing with the formation of Py-DAB and PyN-DAB (Supplementary Fig. 23). Meanwhile, the holes and electrons in Py-DAB and PyN-DAB are significantly separated (Supplementary Figs. 24, 25), which promotes the occurrence of electron transfer at the interface between the donor and acceptor. The above results show that our synthesized propenone-linked COFs have the advantage of superior electro-optical activity and are expected to be potential candidates in environment filed.

## Adsorption and photocatalysis of uranium investigations

Uranium is a key element related to the sustainable development of the nuclear industry[31], but uranium is also a global environmental pollutant with combined radio and chemo toxicity[32–34]. In recent years, various sorbents have been developed for uranium removal[6,35,36], Among them, amidoxime-based COFs are widely used for uranium extraction due to their abundant binding sites, regular pore structure and large specific surface area[37], but the process of amidoxime always leads unavoidable vanadium interference[21]. The uniformly distributed C=O and pyridine nitrogen groups can specifically complex with uranium, which can overcome the shortcomings of amidoxime-based adsorbents[22,25]. Meanwhile, photocatalytic reduction of soluble uranium (VI) to insoluble uranium (IV) has been proposed as a green and viable solution to the health and environmental problems caused by uranium-containing wastewater[38].

The *sp²-c* conjugated Py-DAB and PyN-DAB were used to photocatalytic uranium reduction. Firstly, the cyclic voltammetry curves of Py-DAB and PyN-DAB were tested, and the onset oxidation potentials

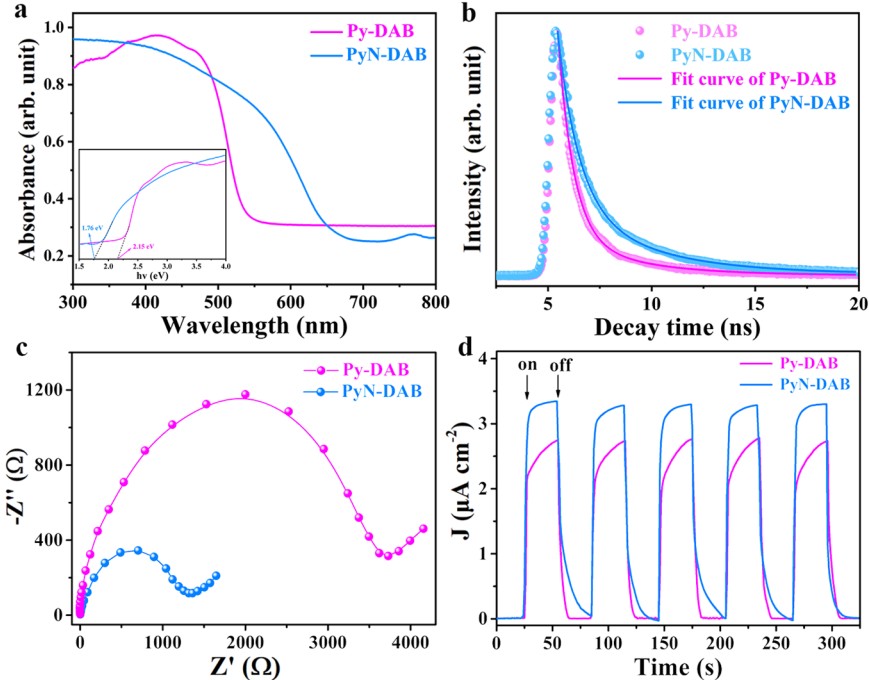

**Fig. 4 | Optical and electronic properties. a** UV-vis diffuse reflection spectrum of Py-DAB and PyN-DAB (Inset: the optical band gaps of Py-DAB and PyN-DAB according to the Kudelka-Munk-transformed reflectance spectra). **b** PL lifetime decay spectra of Py-DAB and PyN-DAB. **c** EIS curves of Py-DAB and PyN-DAB. **d** Photocurrent generation test spectrum of Py-DAB and PyN-DAB.

of Py-DAB and PyN-DAB were about −1.15 V and −1.28 V (Supplementary Figs. 26, 27), respectively, which is much lower than the reduction potential 0.411 V of $UO_2^{2+}/UO_2^{39}$, indicating that the reduction of $UO_2^{2+}$ by Py-DAB and PyN-DAB was feasible. However, the imine-linked Py-PD had no obvious redox peak (Supplementary Fig. 28), proving that the introduction of propenone structure can endow Py-DAB and PyN-DAB with reducibility. Then, we optimized the pH value at 1.0–5.0, and obtained the maximum capture capacity (711.9 mg g$^{-1}$ and 1436.4 mg g$^{-1}$, respectively) under the conditions of pH 4.0 (Fig. 5a), the decrease of adsorption capacity at pH 5.0 was attributed to the existence of multiple forms of uranium[6]. PyN-DAB had the higher adsorption capacity compared to Py-DAB may be attributed to the pyridine nitrogen and the relatively closer oxygen atom can specifically bind to uranium (Supplementary Fig. 29)[25,39]. Even at pH 1.0, the uranium capture capacity of PyN-DAB (418.4 mg g$^{-1}$) was still higher than that of COF-PDAN-AO (105 mg g$^{-1}$)[37]. In addition, the test results of the Zeta potentials proved that both Py-DAB and PyN-DAB have more negative potential compared to the amine-linked Py-PD, indicating that the introduction of propenone structure can increase the electronegativity of materials skeleton (Supplementary Fig. 30), which is beneficial for promoting the adsorption of $UO_2^{2+}$.

The behaviors of Py-DAB and PyN-DAB in photocatalytic reduction of uranium were systematically tested. Clearly, the uranium adsorption capacity of PyN-DAB obviously increased from 415.8 (dark) to 1436.4 mg g$^{-1}$ (light irradiation) (3.45-fold) (Fig. 5b), which was obviously higher to the majority of previously reported porous framework adsorbents (Supplementary Table 5). In addition, kinetic experiments on the adsorption of $UO_2^{2+}$ were carried out at the same initial concentration of $UO_2^{2+}$ and different reaction times. The results showed that the uranium adsorption capacity of PyN-DAB reached adsorption equilibrium at around 60 min and faster than Py-DAB due to the abundant specific sites (Fig. 5c), which is fitted with the curve of Pseudo-second-order model[40]. PyN-DAB still kept remarkable removal rate (>90%) after 5 cycles (Supplementary Fig. 31). It was also clear from the FTIR spectrum that PyN-DAB had well adsorption-desorption properties (Supplementary Fig. 32), and the excellent retention of

PXRD and BET surface areas after desorption proved that PyN-DAB had good stability (Supplementary Figs. 33, 34).

In order to explore the application value of PyN-DAB in the actual environment, we conducted anti-interference tests under a variety of conditions. Uranium removal efficiency of PyN-DAB in the presence of competitive cations and the simulated nuclear industry wastewater ($UO_2^{2+}$ concentration is about 28.76 ppm, Supplementary Table 6) was tested[41], and the results showed that all removal rates were >90% under visible light irradiation (Fig. 5d). Particularly, PyN-DAB was applied in four kinds of uranium mine wastewater (Supplementary Tables 7–10), and the test results all showed ultra-high removal efficiency (>98%), which is proved that PyN-DAB had excellent application value in the real environment. To further evaluate the practical application, *P. aeruginosa*, *S. aureus* and *V. alginolyticus* were used as model samples to test the bacterial activity, and the results indicated that PyN-DAB had broad-spectrum anti-biosiltation activity (Supplementary Fig. 35).

**Research on mechanism of photocatalytic uranium reduction**

To research the interaction between Py-DAB, PyN-DAB and $UO_2^{2+}$, the structural varied of Py-DAB and PyN-DAB before and after binding with uranium was analyzed by X-ray photoelectron spectroscopy (XPS). The U 4$f$ peaks were more pronounced after uranium adsorption in the light than the dark (Fig. 6a and Supplementary Fig. 36), indicating that uranium was successfully captured in PyN-DAB and that light was able to greatly increase uranium adsorption capacity. In addition, the binding energy of O 1$s$ or N 1$s$ was obviously shifted towards a higher energy field after uranium absorption (dark and light), and even the O-U peak appeared (Fig. 6b and Supplementary Figs. 37–42)[6,42]. Furthermore, the peaks of U 4$f^{7/2}$ (382.2 and 380.6 eV) and U 4$f^{5/2}$ (392.8 and 391.4 eV) coexisted under visible light irradiation, and the peaks area of U$^{IV}$ is much larger than that of U$^{VI}$ (Fig. 6c and Supplementary Fig. 43), proving that U$^{VI}$ was reduced to U$^{IV}$ during the sorption of PyN-DAB[43,44]. Compared to the actual situation where U$^{VI}$ is diamagnetic and electron paramagnetic resonance (EPR) silent[45], PyN-DAB had strong EPR signal with $g$ = 1.99 under visible light irradiation (Fig. 6d), consistent with the g value of

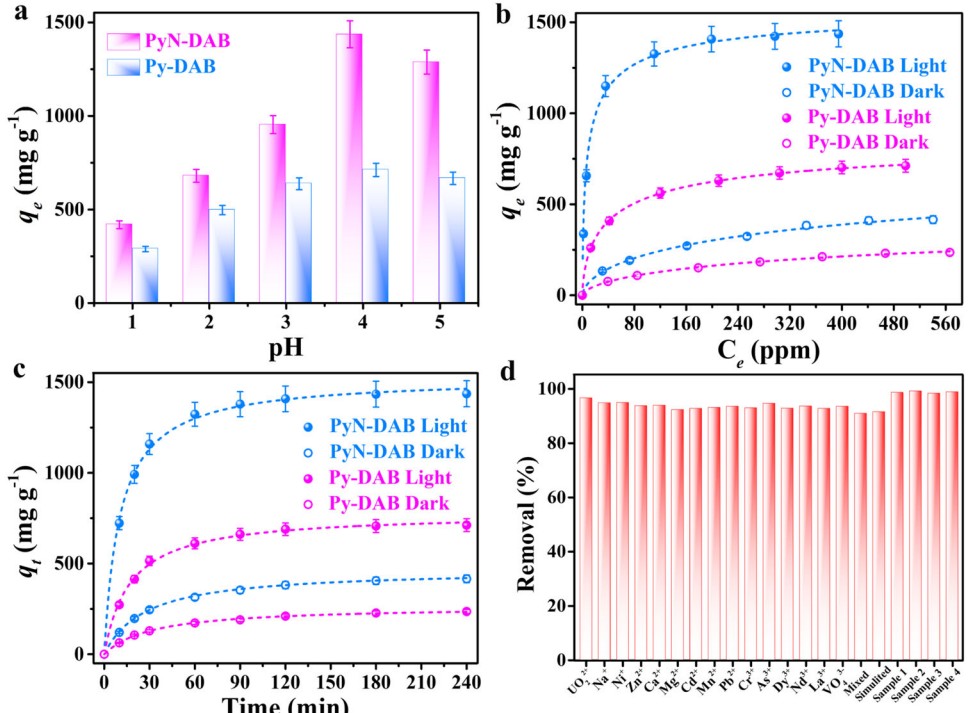

**Fig. 5 | Adsorption and photocatalysis of uranium. a** Under visible light irradiation, the uranium adsorption capacity of PyN-DAB at pH 1.0–5.0. **b** The adsorption isotherm of $UO_2^{2+}$ on PyN-DAB under dark and light irradiation (pH 4.0), the dotted lines are the fitting curve of Langmuir model. **c** The $UO_2^{2+}$ adsorption kinetics on PyN-DAB under dark and light irradiation (pH 4.0). **d** The uranium removal rate of PyN-DAB in the presence of competitive cations (initial concentrations are 50 ppm), mixed solution, simulated solutions and four uranium mine wastewater samples. Error bars represent S.D. $n$ = 3 independent experiments.

$U^{IV}$, which further confirmed that PyN-DAB had excellent photocatalytic reduction performance from $U^{VI}$ to $U^{IV}$.

To confirm the above combined mechanism, the typical molecular structure containing adsorption sites (named C and N, respectively) were selected to perform quantum-chemical calculations by the DFT method. The surface charge density distribution results showed that carbonyl oxygen and pyrazine nitrogen atoms exhibited higher electron cloud density (Fig. 6e and Supplementary Fig. 44). In sharp contrast, aromatic ring carbon atoms had little charge accumulation. Therefore, the electron delocalization region between carbonyl oxygen and pyridine nitrogen atoms was considered to be the main interaction region. To further explore the specific coordination behavior of uranium in Py-DAB and PyN-DAB, their binding energy with uranium was calculated. The results of the theoretically calculated binding energy are about −107.4 and −346.6 kJ mol⁻¹, respectively (Fig. 6f and Supplementary Fig. 45), proving that N can more effectively combine uranium.

## Discussion

In summary, we have successfully demonstrated a feasible approach to construct propenone-linked COFs via Claisen-Schmidt condensation reaction. Due to the introduction of propenone structure, Py-DAB and PyN-DAB revealed high crystallinity, excellent physicochemical stability, and wonderful photocatalytic functions. Furthermore, application of them to the photocatalytic reduction of uranium showed that PyN-DAB exhibited stronger adsorption capacity and faster adsorption kinetics for uranium than Py-DAB due to its abundant binding sites and broader negative charge region. It is worth highlighting that PyN-DAB synthesized by the one-pot method showed excellent removal rates (>98%) in four uranium-containing wastewater samples, avoiding the use of toxic cyano monomers and significantly overcoming the drawbacks of amidoxime-based COFs in complex post-modification

processes and relatively poor selectivity. This ingenious protocol expands the synthetic toolbox of $sp^2$ carbon-linked COFs, which can stimulate the research on enriching the reaction types and monomers availability, and further explores the broad application prospects of COFs.

## Methods
### Synthesis of Py-DAB
A mixture of TFPPy (18.6 mg), DAB (9.7 mg) and 1,4-dioxane solution (1.5 mL) in a Pyrex tube with aqueous KOH (0.25 mL, 4 M) was degassed by three freeze-pump-thaw cycles. The tube was sealed off and heated at 90 °C for 72 h. The yellow solid was filtrated and then washed with $H_2O$, tetrahydrofuran (THF), N,N-Dimethylformamide (DMF) and dimethyl sulfoxide (DMSO), and dried under vacuum at 100 °C to obtain Py-DAB solid of 89% yield.

### Synthesis of PyN-DAB
A mixture of TFPPyN (18.6 mg), DAB (9.7 mg) and 1,4-dioxane solution (1.5 mL) in a Pyrex tube with aqueous KOH (0.25 mL, 4 M) was degassed by three freeze-pump-thaw cycles. Then the tube was sealed off and heated at 90 °C for 72 h. The yellow solid was filtrated and washed with $H_2O$, THF, DMF and DMSO, and dried under vacuum at 100 °C to obtain PyN-DAB solid of 75% yield.

### Synthesis of Py-PD
A mixture of TFPPy (24.7 mg), p-Phenylenediamine (8.7 mg), 1,4-dioxane (0.8 mL), mesitylene (1.334 mL), benzyl alcohol (0.8 mL) and acetic acid solution (0.2 mL, 6 M) was degassed by three freeze-pump-thaw cycles. Then the tube was sealed off and heated at 120 °C for 72 h. The yellow solid was filtrated and washed with $H_2O$, THF, DMF and DMSO, and dried under vacuum at 80 °C to obtain Py-PD solid of 93% yield.

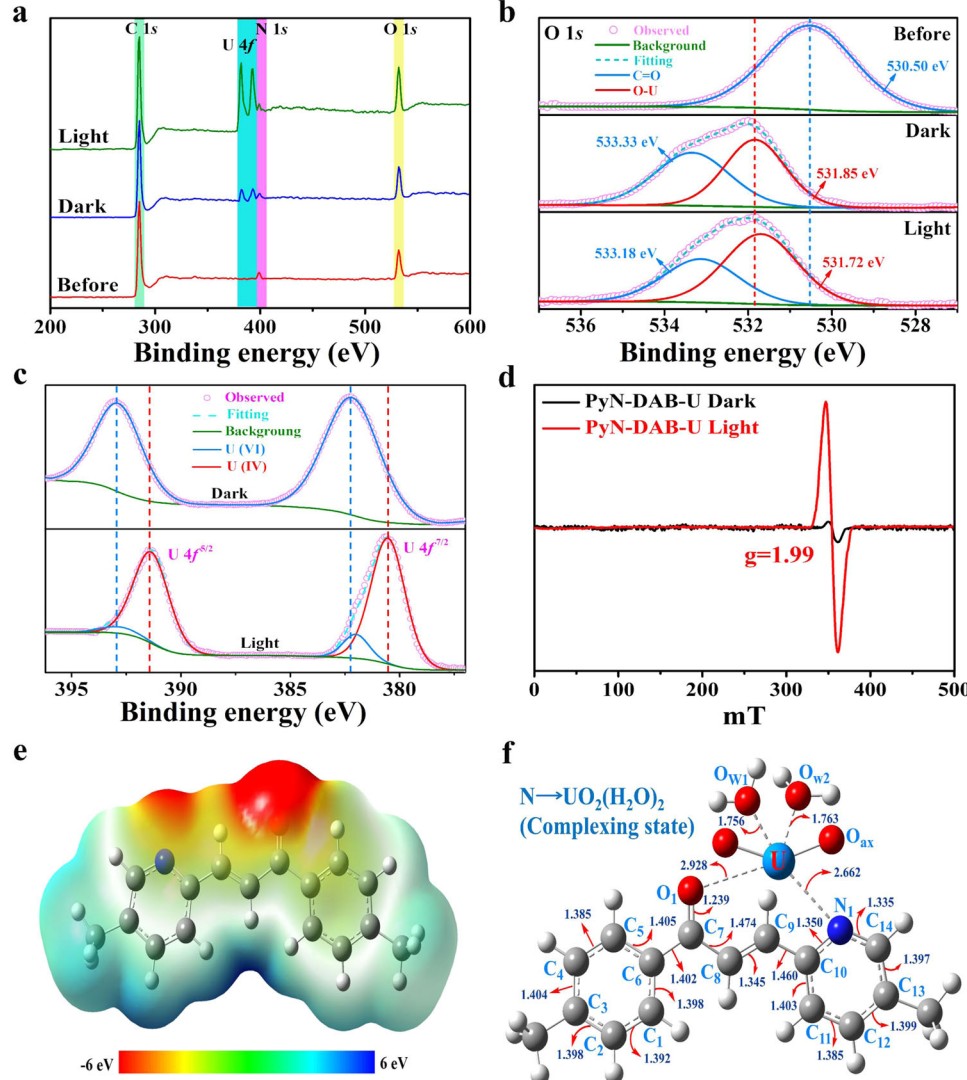

**Fig. 6 | The interactional mechanism between PyN-DAB and uranium. a** XPS survey spectra of PyN-DAB measured before and after loaded with uranium in the dark or visible light irradiation. **b** High-resolution XPS spectra of O 1s for PyN-DAB. **c** U 4f high-resolution spectra of PyN-DAB loaded with uranium in the dark or visible light irradiation. **d** EPR spectra of PyN-DAB loaded with uranium in the dark or visible light irradiation. **e** Corresponding electrostatic surface potential distribution of N. **f** The optimized structures for the complexing state of uranium with N.

## Data availability

The data supporting the findings of this study are available in the article and Supplementary Information files or available from the corresponding authors upon request.

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

## Acknowledgements

We gratefully acknowledge the supports from the National Natural Science Foundation of China (No. 22036003 (J.-D.Q), 21976077 (J.-D.Q.) and 22176082 (R.-P.L.)) and Natural Science Foundation of Jiangxi Province (No. 20212ACB203009 (J.-D.Q.) and 20212ACB203011 (R.-P.L.)).

## Author contributions

J.-D.Q., C.-P.N. and C.-R.Z. conceived and designed the research. C.-P.N. performed the synthesis and conducted the experiments, C.-R.Z. performed the characterizations. X.L. conducted the theoretical calculations. R.-P.L. and J.-D.Q supervised the project and discussed the experiments. C.-P.N., C.-R.Z., R.-P.L. and J.-D.Q. participated in drafting the paper and gave approval to the final version of the manuscript.

## Competing interests

The authors declare no competing interests.
