## [Peer Review File · Nature Communications]

Synthesis of propenone-linked covalent organic frameworks
via Claisen-Schmidt reaction for photocatalytic removal of
uraniumREVIEWER COMMENTS

Reviewer #1 (Remarks to the Author):

In this manuscript, the authors reported a novel strategy for the preparation of sp²-c conjugated covalent organic frameworks (COFs). Two COFs synthesized in this work innovatively introduce propenone structure as connection, which endows them with excellent stability and photoelectric properties, and significantly enriches the reaction types and monomers availability. The authors have also provided great figures and reasonable explanations to support their findings, especially the realization of efficient purification of uranium-containing wastewater in real environment, which further proves the practical value of COFs in the field of environmental protection. I believe the research is interesting and comprehensive, therefore, I think that this manuscript is suitable for publication in Nature Communication after minor revisions addressing the following issue.

Special comments:

1. The authors have claimed as "Py-DAB and PyN-DAB had a staggered A-B eclipsed stacking structure". However, the authors not provide comparison data for the A-A stacking model in the whole manuscript, please supplement the comparison of A-A stacking model and explain the possible reasons for forming A-B stack.
2. Both the synthesized Py-DAB and PyN-DAB in this work have high crystallinity and clear lattice fringes by PXRD and HR-TEM analysis. Why is BET only at a moderate level? Please provide a reasonable explanation.
3. From the author's experimental optimization process for capturing uranium, it can be found that the impact of pH is very significant. In particular, the capture ability decreases severely at low pH. What is the reason for this result? In addition, what are the pH values of the four real uranium-containing wastewater used in this work? Please provide a reasonable explanation and supplement relevant information.
4. The materials in this manuscript did not involve amidoxime groups, while the authors emphasized that amidoxime-based COFs have the poor selectivity produced by vanadium. Please explain why.
5. The authors have done a lot of tests on materials and their properties, but a few details are still neglected, such as the specific operating steps of the cycle test and the explanation of elution mechanism. Please supplement the corresponding details.

6. Some language forms and graphic details in the manuscript need to be carefully checked and corrected, such as "is" in line 18 (P18), and "1.70 V" in Figure 3d.
7. Two relevant important papers are helpful to improve the discussion of mechanism interaction such as: [Doi.org/10.1038/s41467-023-36710-x](https://doi.org/10.1038/s41467-023-36710-x); [Doi.org/10.1021/jacsau.2c00614](https://doi.org/10.1021/jacsau.2c00614).
8. Please supplement a reasonable TOC diagram.

Reviewer #2 (Remarks to the Author):

In this manuscript, the authors investigated a new linkage to design COFs, Propenone-Linked COFs, as a possible catalyst for photocatalytic removal of uranium. The propenone-Linked COFs with high physicochemical stability, crystallinity and extend π -conjugated skeleton, which opened a new door for the design of building blocks bearing new functional groups. In addition, these COFs showed excellent removal rates (>98%) in four uranium mine wastewater samples, especially PyN-DAB. However, some concerns should be addressed before this manuscript can be reconsidered for publication in this journal.

1, Please double-check the supporting information, there are a lot of important figures missing.

2, It is noted that your manuscript needs careful editing by someone with expertise in technical English editing paying particular attention to English grammar, spelling, and sentence structure so that the goals and results of the study are clear to the reader.

3, What does COFs stand for? Covalent Organic Frameworks, I presume, but it is only defined on line 24 for the first time.

4, "In addition to showing a crystallinity comparable to their counterparts prepared by the widely used Knoevenagel reaction, the monomers used in this strategy are also more in line with the concept of green chemistry." Why you say that the monomers used in this strategy are also more in line with the concept of green chemistry? How to prove?

5, "Furthermore, the synthesized Py-DAB and PyN-DAB exhibited excellent physicochemical stability and photocatalytic performance due to the introduction of the propenone structure." How to prove that they are caused by the introduction of the propenone structure?

6, What does Py-DAB stand for on the line 97?

7, "As shown in Figure 1a and 1b, Py-DAB and PyN-DAB had a staggered A-B eclipsed stacking structure with high crystallinity at low angle," why only at low angle? How about high angle? Could you show the simulated PXRD patterns both of the A-A and A-B stacking structure?

8, "corresponding to the (110) and (020) planes, respectively," which peak corresponding?

9, "Additionally, the Pawley refinement for these COFs yielded PXRD patterns showed reasonable agreement with the simulation", What software was used? Can you show the unweight-profile R factor (R_p) and weight-profile factor (R_{wp}) both of the Py-DAB and PyN-DAB?

10, "and the images confirmed that these two COFs have homogeneous rod-like shape (Figure S5-S8)." Which figures? SEM or TEM?

11, Provide more information on the energy distribution of the reactions in the different states was performed. Please show the structures of cis-trans isomerism, and the pathway of their formation. The data are improperly interpreted to prove that the cis structure this step is a heat-absorbing reaction?

12, How are the peaks in NMR in Figure 2 a and b assigned?

13, Could you explain that why the absorption edge for PyN-DAB is blue-shift by about 100 nm over the Py-DAB?

14, Provide more information on how the PL decay spectral was got.

15, A better way of analysing the data of lifetime has been reported, for example: High-Efficiency Photoenhanced Extraction of Uranium from Natural Seawater by Olefin-Linked Covalent Organic Frameworks.

16, line 194, how to prove that the reducibility of these COFs were caused by the introduction of ketone structure?

17, line 198, why you say that they existed a much more efficient separation of interfacial charge? Where is the interfacial?

18, Provide more information on the DFT was performed. How to identify the electrons and holes? How to prove that the overlap of molecular orbitals was reduced significantly? What is the distance between the electrons and the holes centroid?

19, The data are improperly interpreted on the line 210. The band gap of the material only determines the light absorption range of the photocatalytic materials, can not be used to determine whether it is favourable to electron transfer.

20, line 256, how to prove that which was attributed to the introduction of the propanone structure?

21, line 283, higher? Please read more papers to learn how to describe changes in XPS.

22, Biofouling can degrade the uranium extraction efficiency of adsorbents by passivating adsorption sites, thus reducing its usefulness for practical applications. Did you measurement the antibiofouling activity of these COFs samples?

23, what is the EPR stand for on the line 290?

24, Provide more information on the DFT to confirm the combine mechanism. Why you calculate the binding energy of the C coordination with the U not the O coordination with the U?

Reviewer #3 (Remarks to the Author):

This work developed two novel propanone-linked COFs based on nucleophilic addition reaction of ketone-activated α -H with aromatic aldehydes, which is an original strategy for synthesis of sp²-c conjugated COFs. Interestingly, the authors applied them in the field of photocatalysis and exhibited intriguing properties, especially PyN-DAB achieved ultra-high removal efficiency for multiple actual uranium-containing water samples, which was significant to the actual environmental protection. Overall, the design of the work was elegant and the data have been clearly presented. Therefore, I think that this work deserves publication in Nature Communication after a minor revision, and some suggestions are listed as follows:

1. The focus of this work is to develop a new strategy for synthesizing sp²-c conjugated COFs, but the performance of Py-DAB and PyN-DAB in uranium capture, especially the mechanism of photocatalytic reduction of uranium was extensively discussed. Please make corresponding modifications to highlight the focus.
2. The results of bandgaps measured by optical properties using the Kubelka-Monk formula in Figure 3b are very different compared with the bandgaps calculated by DFT in Figure 4, Please provide a reasonable explanation. In addition, Can Figure 4 be merged with Figure 3? Or put Figure 4 in supporting information?
3. As is well known, vanadium often causes strong interference in the uranium removal

process. However, as shown in Figure 5d, vanadium hardly causes interference in the uranium removal process by PyN-DAB. Please explain reason.

4. To prove the introduction of ketone structure can improve the electronegativity of the COFs, the authors selected a repeating unit for DFT theoretical calculation to indicate their surface electrostatic distribution. Zeta potential should also be added.

5. Some graphic details need to be checked and corrected, such as the deformation of “N₂ uptake cm³ g⁻¹” in Figure 2d, and the horizontal axis in Figure 5a. Meanwhile, the type represented by the color of the atom should be indicated in Figure 1a and 1b.

Response to Reviewers (NCOMMS-23-10968)

The authors are very grateful to the reviewers for their valuable comments. These comments are valuable and very helpful for revising and improving our paper, as well as the important guiding significance to our researches. Here we provide a detailed point-by-point response to the Reviewers' comments, and we have edited the Main Manuscript and the Supplementary Information accordingly.

Reviewer #1

In this manuscript, the authors reported a novel strategy for the preparation of sp^2 - c conjugated covalent organic frameworks (COFs). Two COFs synthesized in this work innovatively introduce propenone structure as connection, which endows them with excellent stability and photoelectric properties, and significantly enriches the reaction types and monomers availability. The authors have also provided great figures and reasonable explanations to support their findings, especially the realization of efficient purification of uranium-containing wastewater in real environment, which further proves the practical value of COFs in the field of environmental protection. I believe the research is interesting and comprehensive, therefore, I think that this manuscript is suitable for publication in Nature Communication after minor revisions addressing the following issue.

1. The authors have claimed as "Py-DAB and PyN-DAB had a staggered A-B eclipsed stacking structure". However, the authors not provide comparison data for the A-A stacking model in the whole manuscript, please supplement the comparison of A-A stacking model and explain the possible reasons for forming A-B stack.

Response: Thanks for the reviewer's valuable suggestions. According to the reviewer's suggestion, the comparison data for the A-A stacking model and the possible reasons for forming A-B stack were supplemented in the Revised Manuscript. "As shown in Figs. 2a and 2b, both Py-DAB and PyN-DAB showed strong diffraction peaks at low angle, which confirmed their long-range ordered framework structure. To verify the stacking mode of COFs, AA and AB stacking model were simulated by Material Studio (Figs. 2a and 2b, Figs. S5 and S6), and used the density functional bonding (DFTB) method to quantitatively evaluate the interlayer interaction. The results showed that AB mode with higher stacking energy ($523.83 \text{ kcal mol}^{-1}$) than AA mode ($428.75 \text{ kcal mol}^{-1}$), proving the staggered AB stacking structure is much more stable.^[17] This may be attributed to the

strong dipole moment generated by the heteroatoms (oxygen atoms) in the skeleton to avoid these charged atoms directly located at the top of each other.^[4] The Pawley refinement for these COFs yielded PXRD patterns showed reasonable agreement with the simulation of AB modes, which was remarkably similar to previous reports.^[18,19] Additionally, Py-DAB with negligible residuals $R_{wp} = 5.58\%$ and $R_p = 2.70\%$, and PyN-DAB with negligible residuals $R_{wp} = 3.84\%$ and $R_p = 1.27\%$, which were attributed to the existence of crystalline 2D network.^[20] (Please see pages 6-7 of the Revised Manuscript, and Figs. S5 and S6 in the Supporting Information).

2. Both the synthesized Py-DAB and PyN-DAB in this work have high crystallinity and clear lattice fringes by PXRD and HR-TEM analysis. Why is BET only at a moderate level? Please provide a reasonable explanation.

Response: Thanks for the reviewer's valuable suggestions. Compared with BET of some reported sp^2 -c conjugated COFs, such as V-2D-COFs ($118\text{-}333\text{ m}^2\text{ g}^{-1}$, *Angew. Chem. Int. Ed.* 2022, 61, e2022097), 2D-PPQV1 ($440\text{ m}^2\text{ g}^{-1}$, *Angew. Chem. Int. Ed.* 2020, 59, 23620), COF-DFB ($1104\text{ m}^2\text{ g}^{-1}$, *J. Am. Chem. Soc.* 2022, 144, 3653), etc., the synthesized Py-DAB and PyN-DAB was at a moderate level, which can be attributed to that under the A-B stacking mode, the pore size is reduced due to the mismatching of pristine pores (*J. Am. Chem Soc.* 2021, 143, 16206).

3. From the author's experimental optimization process for capturing uranium, it can be found that the impact of pH is very significant. In particular, the capture ability decreases severely at low pH. What is the reason for this result? In addition, what are the pH values of the four real uranium-containing wastewater used in this work? Please provide a reasonable explanation and supplement relevant information.

Response: Thanks for the reviewer's valuable suggestions. The effect of pH value on the uranium capture ability of two COFs was predominantly due to the competition between H^+ and U (VI) for adsorption sites (*Chem. Eng. J.* 2019, 365, 231). As the solution pH decreased, i.e., the concentration of H^+ increased, the binding ability of H^+ to the adsorption sites in the framework enhanced, leads to the decreased uranium capture ability. In addition, the pH values of the four real uranium-containing wastewater used in this work are 1.70, 1.65, 3.11 and 8.02, respectively. Due to the excellent photocatalytic reduction ability of PyN-DAB, even at pH 1.0, the uranium capture capacity of PyN-DAB

(418.4 mg g⁻¹) was still higher than COF-PDAN-AO (105 mg g⁻¹) (*J. Hazard. Mater.* 2020, 392, 122333), and the uranium content in four uranium-containing samples is relatively low, so PyN-DAB can achieve ultra-high uranium removal rates (> 98%). In addition, the relevant information has been supplemented in the Supporting Information. (Please see Tables S5-S8 in the Supporting Information).

4. The materials in this manuscript did not involve amidoxime groups, while the authors emphasized that amidoxime-based COFs have the poor selectivity produced by vanadium. Please explain why.

Response: Thanks for the reviewer's valuable suggestions. Amidoxime-based adsorbents have become a popular material for uranium adsorption in recent years. The cyclic imide-dioxime form generated during the amide oximation process has a strong binding ability on vanadium. In addition, the broad pH region where the V(V) complex is stable also helps to explain the difficulty of removing vanadium from amidoxime-based sorbents during the elution process (*Nat. Commun.* 2017, 8, 1560), which seriously affects the performance of this type of adsorbent. Currently, *sp*²-*c* linked COFs adsorbents of uranium were mainly synthesized through Knoevenagel reaction, and then the cyano group was amidoximized through post modification (*Angew. Chem. Int. Ed.* 2020, 59, 17684; *Nat. Commun.* 2020, 11, 436). This process will lead unavoidable vanadium interference. However, the uniformly distributed C=O and pyridine nitrogen groups in the PyN-DAB synthesized in this work can specifically complex with uranium, which can overcome the shortcomings of amidoxime-based adsorbents. Discussions were supplemented in the Revised Manuscript. "Among them, amidoxime-based COFs are widely used for uranium extraction due to their abundant binding sites, regular pore structure and large specific surface area,^[37] but the process of amidoxime always leads unavoidable vanadium interference.^[21] The uniformly distributed C=O and pyridine nitrogen groups can specifically complex with uranium, which can overcome the shortcomings of amidoxime-based adsorbents.^[22,25]" (Please see page 14 of the Revised Manuscript).

5. The authors have done a lot of tests on materials and their properties, but a few details are still neglected, such as the specific operating steps of the cycle test and the explanation of elution mechanism. Please supplement the corresponding details.

Response: Thanks for the reviewer's valuable suggestions. The corresponding details of cycle tests have been supplemented in Supporting Information. "Recyclability test: After one run of adsorption, the adsorbents were regenerated by treatment with the elution solution of 0.1 M HNO₃ solution and reused for another adsorption experiment. For 5 mg adsorbents, a certain amount of elution solution was used to elute the binding uranium at room temperature. The elution efficiency (E, %) was determined by using Equation:

$$E = \frac{C_e \times V_e}{(C_o - C_t) \times V_t} \times 100\%$$

in where C_e (mg L⁻¹) is the uranium concentration in elution solution, V_e (L) is the volume elution solution, C_t (mg L⁻¹) is the uranium concentration in the simulated nuclear industry wastewater after uranium adsorption, C_o (mg L⁻¹) is the initial uranium concentration of the simulated nuclear industry wastewater, V_t (L) is the volume of simulated nuclear industry wastewater used for adsorption. The resulting suspension was filtered and washed with ultra-pure water till the supernatant became neutral. After being dried under vacuum, the resultant material was used for another adsorption experiment. It was found that after five consecutive cycles still showed excellent uranium removal rate." (Please see Recyclability test section of the Supporting Information). In addition, the elution mechanism has been supplemented in the Supporting Information. "In addition, the explanation of elution mechanism is that because of the concentration of H⁺ in the elution solution is much higher than the concentration of H⁺ in the adsorption experiment, a large amount of H⁺ will occupy the binding sites of uranyl on COF during the desorption process, resulting in the successful desorption of uranyl from COF." (Please see the annotations of Fig. S30 in the Supporting Information).

6. Some language forms and graphic details in the manuscript need to be carefully checked and corrected, such as "is" in line 18 (P18), and "1.70 V" in Figure 3d.

Response: Thanks for the reviewer's valuable suggestion. We have carefully checked and corrected the language forms and graphic details in the manuscript (Please see the Revised Manuscript and Supporting Information).

7. Two relevant important papers are helpful to improve the discussion of mechanism interaction such as: [Doi.org/10.1038/s41467-023-36710-x](https://doi.org/10.1038/s41467-023-36710-x); [Doi.org/10.1021/jacsau.2c00614](https://doi.org/10.1021/jacsau.2c00614).

Response: Thanks for the reviewer's valuable suggestion. Two relevant important papers helpful to improve the discussion of mechanism interaction have been added in the Revised Manuscript (Please see references 29 and 40 in the Revised Manuscript).

8. Please supplement a reasonable TOC diagram.

Response: According to the reviewer's suggestion, the TOC diagram has been supplemented in the Revised Manuscript.

Reviewer #2

In this manuscript, the authors investigated a new linkage to design COFs, Propenone-Linked COFs, as a possible catalyst for photocatalytic removal of uranium. The propenone-Linked COFs with high physiochemical stability, crystallinity and extend π -conjugated skeleton, which opened a new door for the design of building blocks bearing new functional groups. In addition, these COFs showed excellent removal rates (>98%) in four uranium mine wastewater samples, especially PyN-DAB. However, some concerns should be addressed before this manuscript can be reconsidered for publication in this journal.

1, Please double-check the supporting information, there are a lot of important figures missing.

Response: Thanks for the reviewer's valuable suggestion. According to the reviewer's suggestion, we have carefully checked the supporting information and supplemented important figures (Please see the Supporting Information).

2, It is noted that your manuscript needs careful editing by someone with expertise in technical English editing paying particular attention to English grammar, spelling, and sentence structure so that the goals and results of the study are clear to the reader.

Response: Thanks for the reviewer's valuable suggestion. According to the reviewer's suggestion, the English grammar, spelling, and sentence structure have been carefully checked and corrected (Please see the Revised Manuscript).

3, What does COFs stand for? Covalent Organic Frameworks, I presume, but it is only defined on line 24 for the first time.

Response: Thanks for the reviewer's valuable suggestion. According to the reviewer's

suggestion, the full name of COFs has been supplemented in the abstract section “The type of reactions and the availability of monomers for the synthesis of sp^2 - c linked covalent organic frameworks (COFs) were considerably limited by the irreversibility of the C=C bond.” (Please see the abstract section in the Revised Manuscript).

4, “In addition to showing a crystallinity comparable to their counterparts prepared by the widely used Knoevenagel reaction, the monomers used in this strategy are also more in line with the concept of green chemistry.” Why you say that the monomers used in this strategy are also more in line with the concept of green chemistry? How to prove?

Response: Thanks for the reviewer’s valuable suggestion. As we all know, the widely used in Knoevenagel reaction required for the monomers must contain aryl α -carbon atoms with sufficient electron defects to enhance the acidity of the α -proton and stabilize the reaction intermediate (*Trends Chem.* 2021, 3, 431), which is usually achieved by introducing electron-withdrawing group such as cyano group. Therefore, these modifications of monomers used in Knoevenagel reaction usually imply tedious preparation procedures and the use of toxic reagents (such as NaCN or CuCN), and significantly limit the available reaction sites of the monomers (*J. Am. Chem. Soc.* 2022, 144, 3653). However, in our work, the monomers used in Claisen-Schmidt reaction are free of cyano and more environmentally friendly. According to the reviewer's suggestion, we have revised the corresponding descriptions in the revised manuscript. (Please see the Revised Manuscript)

5, “Furthermore, the synthesized Py-DAB and PyN-DAB exhibited excellent physicochemical stability and photocatalytic performance due to the introduction of the propenone structure.” How to prove that they are caused by the introduction of the propenone structure?

Response: Thanks for the reviewer’s valuable suggestion. Generally, the olefin linked COFs often have better electron transfer properties compared to the imine-linked COFs (*Nat. Commun.* 2021, 12, 4735; *Angew. Chem. Int. Ed.* 2022, 61, e202111627), and the introduction of ketone structures can enhance light absorption and reducibility (*ACS Appl. Mater. Interfaces* 2021, 13, 27041). In addition, lone pair electron resonance effect of oxygen atom can further improve the stability of materials (*J. Am. Chem. Soc.* 2022, 144, 6821), while the propenone structure happens to be an organic combination of them. To

further prove the synthesized Py-DAB and PyN-DAB with excellent physicochemical stability and photocatalytic performance due to the introduction of the propenone structure, an imine linked COFs (Py-PD) as a comparison was synthesized. The results showed that the propenone-linked Py-DAB and PyN-DAB have more powerful advantages in physicochemical stability, photoelectric response ability and other aspects, proving that they are caused by the introduction of the propenone structure. The relevant discussions have been supplemented in the Revised Manuscript. “To further verify the excellent stability and photoelectric performance caused by the introduction of propenone structure, an imine-linked COF (Py-PD) was synthesized and tested as a comparison, and the results showed that it exhibited poor stability and photoelectric performance compared to Py-DAB and PyN-DAB (Fig. S21).” (Please see page 12 of the Revised Manuscript and Fig. S21 in Supporting Information)

6, What does Py-DAB stand for on the line 97?

Response: Py-DAB stands for the condensed product of Py and DAB, which has been corrected in the Revised Manuscript. “Finally, the crystalline condensation product (Py-DAB) was synthesized under the optimal reaction conditions (more details in the Supporting Information).” (Please see page 5 of the Revised Manuscript)

7, “As shown in Figure 1a and 1b, Py-DAB and PyN-DAB had a staggered A-B eclipsed stacking structure with high crystallinity at low angle,” why only at low angle? How about high angle? Could you show the simulated PXRD patterns both of the A-A and A-B stacking structure?

Response: Thanks for the reviewer’s valuable suggestion. Generally, COFs exhibit strong PXRD diffraction peaks at low angles (less than 10°) to demonstrate the existence of long-range ordered structures. In our work, Py-DAB and PyN-DAB have strong PXRD diffraction peaks at low angles, and the clear lattice fringes shown by HR-TEM images both confirm their long-range ordered structures. In addition, the comparison data for the AA stacking model has been supplemented in the Supporting Information. Meanwhile, the density functional tight binding (DFTB) method was employed to evaluate the interlayer interactions quantitatively. For example, PyN-DAB showed stacking energies of 428.75 and 523.83 kcal mol⁻¹ for the AA and staggered AB modes, respectively. Accordingly, the staggered AB tacking structure is much more stable than the AA stacking

structure (*J. Am. Chem. Soc.* 135, 546). This may be attributed to the presence of strong dipole moments generated by heteroatoms in the skeleton (oxygen atom). Therefore, the same charged oxygen atoms were oriented in different directions to avoid these charged atoms being directly on the top of each other, while satisfying dipole-dipole interactions (*Angew. Chem. Int. Ed.* 2022, 61, e202111627). Additionally, the Pawley refinement for these COFs yielded PXRD patterns showed reasonable agreement with the simulation of AB modes by Materials Studio software, Py-DAB with negligible residuals $R_{wp} = 5.58\%$ and $R_p = 2.70\%$, and PyN-DAB with negligible residuals $R_{wp} = 3.84\%$ and $R_p = 1.27\%$, which were attributed to the existence of crystalline 2D network (*Small* 2021, 17, 2103152). These experimental results have been supplemented in the Revised Manuscript and Supporting Information. “As shown in Figs. 2a and 2b, both Py-DAB and PyN-DAB showed strong diffraction peaks at low angle, which confirmed their long-range ordered framework structure. To verify the stacking mode of COFs, AA and AB stacking model were simulated by Material Studio (Figs. 2a and 2b, Figs. S5 and S6), and used the density functional bonding (DFTB) method to quantitatively evaluate the interlayer interaction. The results showed that AB mode with higher stacking energy ($523.83 \text{ kcal mol}^{-1}$) than AA mode ($428.75 \text{ kcal mol}^{-1}$), proving the staggered AB stacking structure is much more stable.^[17] This may be attributed to the strong dipole moment generated by the heteroatoms (oxygen atoms) in the skeleton to avoid these charged atoms directly located at the top of each other.^[4] The Pawley refinement for these COFs yielded PXRD patterns showed reasonable agreement with the simulation of AB modes, which was similar to previous reports.^[18,19] Additionally, Py-DAB with negligible residuals $R_{wp} = 5.58\%$ and $R_p = 2.70\%$, and PyN-DAB with negligible residuals $R_{wp} = 3.84\%$ and $R_p = 1.27\%$, which were attributed to the existence of crystalline 2D network.^[20]” (Please see pages 6-7 of the Revised Manuscript, and Figs. S5 and S6 in the Supporting Information).

8, “corresponding to the (110) and (020) planes, respectively,” which peak corresponding?

Response: According to the structural model of COF simulation of AB stacking, the peak at 6.75° corresponds to 110 crystal planes, and the peak at 7.55° corresponds to 020 crystal planes. The crystal planes corresponding to the specific diffraction peaks in Figs. 2a and 2b have been marked (Please see Figs. 2a and 2b in the Revised Manuscript).

9, “Additionally, the Pawley refinement for these COFs yielded PXRD patterns showed

reasonable agreement with the simulation”, What software was used? Can you show the unweight-profile R factor (Rp) and weight-profile factor (Rwp) both of the Py-DAB and PyN-DAB?

Response: Thanks for the reviewer’s valuable suggestion. Materials Studio software was used in our work. The unweight-profile R factor (Rp) and weight-profile factor (Rwp) both of the Py-DAB and PyN-DAB were supplemented in the Revised Manuscript. “The Pawley refinement for these COFs yielded PXRD patterns showed reasonable agreement with the simulation of AB modes, which was remarkably similar to previous reports.^[18,19] Additionally, Py-DAB with negligible residuals Rwp = 5.58% and Rp = 2.70%, and PyN-DAB with negligible residuals Rwp = 3.84% and Rp = 1.27%, which were attributed to the existence of crystalline 2D network.^[20]” (Please see page 7 of the Revised Manuscript).

10, “and the images confirmed that these two COFs have homogeneous rod-like shape (Figure S5-S8).” Which figures? SEM or TEM?

Response: The description was modified in the Revised Manuscript. “The morphologies of COFs were characterized by scanning electron microscopy (SEM) (Figs. S7 and S8) and transmission electron microscopy (TEM), and the TEM images confirmed that these two COFs have homogeneous rod-like shape (Figs. S9 and S10).” (Please see page 7 of the Revised Manuscript).

11, Provide more information on the energy distribution of the reactions in the different states was performed. Please show the structures of cis-trans isomerism, and the pathway of their formation. The data are improperly interpreted to prove that the cis structure this step is a heat-absorbing reaction?

Response: Thanks for the reviewer’s valuable suggestion. According to the reviewer’s suggestion, the structures of cis-trans isomerism and the pathway of their formation were revised and provided in the Supporting Information (Figs. S13 and S14). And detailed descriptions were supplemented in the caption of Fig. S14. “**Fig. S14** | DFT calculated molecular conformation and energy profiles of the suggested Claisen-Schmidt leading to the trans -C=C- bond under catalysis of NaOH. In step 1, the 1,4-diacetophenone I tends to be easily deprotonated by OH⁻ to form the relatively low energy carbon anion species II (E=0.05 kJ/mol). Afterward, the carbanion species II attacks the aldehyde to yield the intermediate III (E=154.7 kJ/mol). Subsequent formation of an intermediate IV (E=-342.4

kJ/mol) due to thermodynamic advantages and the irreversible elimination of stable result in the trans-vinylene ($E=-708.4$ kJ/mol).” (Please see Figs. S13 and S14 in the Supporting Information).

12, How are the peaks in NMR in Figure 2 a and b assigned?

Response: Thanks for the reviewer’s valuable suggestion. To make the assignation of NMR peaks clearer, the ^{13}C NMR spectra of Py-DAB was compared with the model compound. A well-resolved peak located at about 144 ppm belonged to the newly formed olefin groups (signed as b), the signal at around 135 ppm (signed as a) was typically arising from the olefin-linked phenyl carbon (*J. Am. Chem. Soc.* 2019, 141, 14272), and the peak at 190 ppm was ascribed to the carbon atoms in the ketone groups (d) (*Chem. Commun.* 2020, 56, 880). Other characteristic peaks can also match well, proving the successful synthesis of propenone-linked Py-DAB. Similarly, PyN-DAB also exhibited distinct characteristic peaks. The detailed descriptions have been supplemented in the Revised Manuscript. “As shown in Figs. 3a, a well-resolved peak located at about 144 ppm belonged to the newly formed olefin groups, the signal at around 135 ppm was typically arising from olefin-linked phenyl carbon,^[24] and the peak located at around 190 ppm was ascribed to the carbon atoms in the ketone groups.^[25] Similarly, PyN-DAB also exhibited distinct characteristic peaks (Fig. 3b). The above results proved that Py-DAB and PyN-DAB were successfully synthesized.” (Please see page 9 of the Revised Manuscript).

The comparison of ^{13}C NMR spectra between the Py-DAB and the model compound.

13, Could you explain that why the absorption edge for PyN-DAB is blue-shift by about 100 nm over the Py-DAB?

Response: Thanks for the reviewer's valuable suggestion. Strong electronegative heteroatoms (such as nitrogen atoms) have lone pair electrons, which not only bring about active sites, but also form p- π conjugation with π electrons in COFs, allowing rapid electron transfer and showing excellent photoelectric properties (*Chem. Rev.* 2020, 120, 9363; *J. Electroanal. Chem.* 2022, 923, 116831), which is why the absorption edge for PyN-DAB occurred wider absorption. Furthermore, the relevant discussions have been supplemented in the Revised Manuscript. "The absorption edge of PyN-DAB with wider absorption may be attributed to the introduce of strong electronegative nitrogen atoms, which forms the effect of p- π conjugation with π electrons in COFs.^[27]" (Please see page 11 of the Revised Manuscript).

14, Provide more information on how the PL decay spectral was got.

Response: Thanks for the reviewer's valuable suggestion. More information on how the PL decay spectral was got has been provided in the Supporting Information. "Steady-state photoluminescence (PL) decay spectra were measured at room temperature using FLS 1000 spectrometer (Edinburgh Instruments, UK)." (Please see page S2 of the Supporting Information).

15, A better way of analysing the data of lifetime has been reported, for example: High-Efficiency Photoenhanced Extraction of Uranium from Natural Seawater by Olefin-Linked Covalent Organic Frameworks.

Response: According to the reviewer's suggestion, more comprehensive descriptions have been supplemented in the Revised Manuscript. "The fluorescence lifetimes of Py-DAB and PyN-DAB were measured to confirm the importance of propenone structure in promoting the separation and transportation of photogenerated carriers. The results of the average lifetimes were calculated to be 2.21 and 3.40 ns, respectively (Fig. 4b). The longer lifetime of Py-DAB and PyN-DAB demonstrated that the recombination of photogenerated carriers was further inhibited,^[28,29] which was related to its more conjugated sp^2 -c skeleton." (Please see page 11 of the Revised Manuscript).

16, line 194, how to prove that the reducibility of these COFs were caused by the introduction of ketone structure?

Response: Thanks for the reviewer's valuable suggestion. The carbonyl carbon atom in

ketone structure is sp^2 hybrid, and the three sp^2 hybrid orbitals of carbon atom form three bonds with the connected oxygen atom and carbon atom, respectively. The bond angle is about 120° , which is a plane triangle structure. Due to the high electronegativity of the oxygen atom, the electrons between the carbon-oxygen double bond strongly favor the side of the oxygen atom, so that the carbonyl oxygen atom has partial negative charge and has certain reducing ability. In our work, the initial oxidation potential of Py-DAB and PyN-DAB were tested by cyclic voltammetry. As shown in Figs. S25 and S26, the onset oxidation potentials of Py-DAB and PyN-DAB were about -1.15 V and -1.28 V, respectively, which is much lower than the reduction potential 0.411 V of UO_2^{2+}/UO_2 (*Angew. Chem. Int. Ed.* 2020, 59, 4168), indicating that the reduction of UO_2^{2+} by Py-DAB and PyN-DAB was feasible. In addition, an imine-linked COF (Py-PD) was synthesized for comparison, but there is no reducibility according to the CV test results of Py-PD (Fig. S27). Therefore, the reducibility of these COFs was caused by the introduction of propenone structure. Detailed discussions have been supplemented in the Revised Manuscript. “Firstly, the cyclic voltammetry curves of Py-DAB and PyN-DAB were tested, and the onset oxidation potentials of Py-DAB and PyN-DAB were about -1.15 V and -1.28 V (Figs. S25 and S26), respectively, which is much lower than the reduction potential 0.411 V of UO_2^{2+}/UO_2 ,^[39] indicating that the reduction of UO_2^{2+} by Py-DAB and PyN-DAB was feasible. However, the imine-linked Py-PD had no obvious redox peak (Fig. S27), proving that the introduction of propenone structure can endow Py-DAB and PyN-DAB with reducibility.” (Please see page 14 of the Revised Manuscript, and Figs. S25-S27 in the Supporting Information).

17, line 198, why you say that they existed a much more efficient separation of interfacial charge? Where is the interfacial?

Response: Thanks for the reviewer’s valuable suggestion. Firstly, sp^2 -*c* linked COFs often have better electron transfer properties compared to imine-linked COFs (*Nat. Commun.* 2021, 12, 4735; *Angew. Chem. Int. Ed.* 2022, 61, e202111627), and the introduction of ketone structures can enhance light absorption and reducibility (*ACS Appl. Mater. Interfaces* 2021, 13, 27041). Therefore, the introduction of propenone structure gives them better photo generated electron ability. In our work, theoretical calculations of the separation situation of the electron from the hole in Py-DAB and PyN-DAB were supplemented in the Supporting Information (Figs. S23 and S24). And it can be seen from

Figs. S23 and S24 that the holes and electrons in Py-DAB and PyN-DAB are significantly separated (blocky areas of green represents electrons cloud, blue represents holes), which promotes the occurrence of charge transfer at the interface between the donor and acceptor. Furthermore, their photocurrent intensities are higher compared to the imine-linked Py-PD, therefore they existed a much more efficient separation of interfacial charge. Meanwhile, the longer lifetimes for Py-DAB and PyN-DAB indicated these particular COFs offered better interfacial charge separation and migration in their extended conjugated skeletons, which was beneficial for improving their photocatalytic performance (*Nat. Commun.* 2023, 14, 1106). Detailed discussions have also been supplemented in the Revised Manuscript and Supporting Information. “What's more, compared with the monomers, the recombined molecules have a smaller energy level bandgap, indicated that as the materials further develop from building blocks to extended π conjugated planar structures, the electron transfer process becomes smoother, and the overlapping motion of electrons through proton clouds becomes freer. It can be seen that the holes and electrons in Py-DAB and PyN-DAB are significantly separated from the theoretical calculation results (Figs. S23 and S24), and the distances between the electrons and the holes centroid are 2.526 and 1.998 Å, respectively, which promotes the occurrence of charge transfer at the interface between the donor and acceptor.” (Please see page 13 of the Revised Manuscript, and Figs. S23 and S24 in the Supporting Information).

18, Provide more information on the DFT was performed. How to identify the electrons and holes? How to prove that the overlap of molecular orbitals was reduced significantly? What is the distance between the electrons and the holes centroid?

Response: Thanks for the reviewer’s valuable suggestion. Firstly, theoretical calculations were conducted to determine the separation situation of the electron from the hole, and the results showed significant separation of holes and electrons in Py-DAB and PyN-DAB (green represents electrons, blue represents holes). In addition, detailed descriptions of the overlap of molecular orbitals were supplemented in the Revised Manuscript. “The effectively separated electrons and holes proved that the HOMOs and LUMOs of these molecules were appropriate pairs for electrons transfer.^[3] What's more, compared with the monomers, the recombined molecules have a smaller energy level bandgap, indicated that as the materials further develop from building blocks to extended π conjugated planar structures, the electron transfer process becomes smoother, and the overlapping motion

of electrons through proton clouds becomes freer. It can be seen that the holes and electrons in Py-DAB and PyN-DAB are significantly separated from the theoretical calculation results (Figs. S23 and S24), and the distances between the electrons and the holes centroid are 2.526 and 1.998 Å, respectively, which promotes the occurrence of charge transfer at the interface between the donor and acceptor. Therefore, it can be inferred that the process of electron transfer was more favorable to occur in the whole COFs.^[3] (Please see page 13 of the Revised Manuscript).

The calculated distances between the electrons and the holes centroid are 2.526 and 1.998 Å for Py-DAB and PyN-DAB, respectively. The D index formula was used to measure the distance between holes and the center of mass of electrons:

$$D_x = |X_{\text{ele}} - X_{\text{hole}}| \quad D_y = |Y_{\text{ele}} - Y_{\text{hole}}| \quad D_z = |Z_{\text{ele}} - Z_{\text{hole}}|$$
$$D \text{ index} = \sqrt{(D_x)^2 + (D_y)^2 + (D_z)^2}$$

19, The data are improperly interpreted on the line 210. The band gap of the material only determines the light absorption range of the photocatalytic materials, cannot be used to determine whether it is favorable to electron transfer.

Response: Thanks for the reviewer's valuable suggestion. Compared to monomers, Py-DAB and PyN-DAB have narrower HOMO-LUMO level bandgaps, indicated that as the materials are further developed from building blocks to extended π conjugated planar structures, the electron transfer process becomes smoother, and the overlapping motion of electrons through proton clouds becomes freer. Therefore, the band gap of COFs is significantly reduced compared to monomers, which is consistent with previous reports (*Angew. Chem. Int. Ed.* 2019, 58, 12065; *J. Am. Chem. Soc.* 2019, 141, 14272; *Appl. Catal. B-Environ.* 2021, 294, 120250). The description has been revised in the Revised Manuscript. "What's more, compared with the monomers, the recombined molecules have a smaller energy level bandgap, indicated that as the materials further develop from building blocks to extended π conjugated planar structures, the electron transfer process becomes smoother, and the overlapping motion of electrons through proton clouds becomes freer." (Please see the page 13 of the Revised Manuscript).

20, line 256, how to prove that which was attributed to the introduction of the propanone structure?

Response: Thanks for the reviewer's valuable suggestion. COFs with large surface area

and tunable regular porous structure might be ideal for capturing highly mobile U^{VI} . At present, a large number of COFs based on dynamic imine bonds have been explored for capturing uranium (*Adv. Mater.* 2018, 30, 1705479; *Adv. Sci.* 2019, 6, 1900547). However, their major linkages exhibit poor π -electron communication over the framework and are susceptible to visible light, acids, bases and irradiation, which severely limited their photocatalytic activity, reusability and practical utilization (*Angew. Chem. Int. Ed.* 2019, 58, 14865; *Angew. Chem. Int. Ed.* 2019, 58, 13753). The formation of sp^2 - c linkages is an effective and practical method to construct COFs with extended planar π -conjugation and high stability under severe conditions (*J. Am. Chem. Soc.* 2019, 141, 6848). In our work, Py-DAB and PyN-DAB are linked by propenone (C=C-C=O), the lone pair electron resonance effect of oxygen atoms can further improve the stability of the material (*J. Am. Chem. Soc.* 2022, 144, 6821). To verify this, an imine linked COFs (Py-PD) was synthesized as a comparison and tested the physicochemical stability. The results showed that the imine-linked Py-PD exhibited poor stability and photoelectric performance compared to Py-DAB and PyN-DAB. Therefore, the excellent performances of Py-DAB and PyN-DAB can be attributed to the introduction of the propenone structure. Detailed discussions were supplemented in the Revised Manuscript. “To further verify the excellent stability and photoelectric performance caused by the introduction of propenone structure, an imine-linked COF (Py-PD) was synthesized and tested as a comparison, and the results showed that it exhibited poor stability and photoelectric performance compared to Py-DAB and PyN-DAB (Fig. S21).” (Please see page 12 of the Revised Manuscript and Fig. S21 in the Supporting Information).

21, line 283, higher? Please read more papers to learn how to describe changes in XPS.

Response: Thanks for the reviewer’s valuable suggestion. It was revised in the Revised Manuscript. “In addition, the binding energy of O 1s or N 1s was obviously shifted towards a higher energy field after uranium absorption (dark and light), and even the O-U peak appeared (Fig. 6b and Figs. S36-41).^{[6,42]”} (Please see page 17 of the Revised Manuscript).

22, Biofouling can degrade the uranium extraction efficiency of adsorbents by passivating adsorption sites, thus reducing its usefulness for practical applications. Did you measurement the antibiofouling activity of these COFs samples?

Response: Thanks for the reviewer's valuable suggestion. According to the reviewer's suggestion, the antibiofouling activity of these COFs samples was measured and results were supplemented in the Revised Manuscript. "To further evaluate the practical application, *P. aeruginosa*, *S. aureus* and *V. alginolyticus* were used as model samples to test the bacterial activity, and the results indicated that PyN-DAB had broad-spectrum anti-biosiltation activity (Fig. S34)." (Please see page 17 of the Revised Manuscript and Fig. S34 in the Supporting Information).

23, what is the EPR stand for on the line 290?

Response: The EPR stands for electron paramagnetic resonance, and the full name of EPR has been supplemented in the Revised Manuscript. "Compared to the actual situation where U^{VI} is diamagnetic and electron paramagnetic resonance (EPR) silent" (Please see page 18 of the Revised Manuscript).

24, Provide more information on the DFT to confirm the combine mechanism. Why you calculate the binding energy of the C coordination with the U not the O coordination with the U?

Response: Thanks for the reviewer's valuable suggestion. To confirm the combined mechanism, the typical molecular structure containing adsorption sites (named C and N, respectively) were selected to perform quantum-chemical calculations by the DFT method. And more information on the DFT to confirm the combine mechanism has been provided in the Supporting Information (Please see page S5). In addition, the C used here represents the typical molecular structure containing the adsorption site of Py-DAB rather than a carbon atom, and the N also represents the typical molecular structure containing the adsorption site of PyN-DAB rather than a nitrogen atom. C and N used here were defined in the Revised Manuscript. "To confirm the above combine mechanism, the typical molecular structure containing adsorption sites (named C and N, respectively) were selected to perform quantum-chemical calculations by the DFT method." (Please see page 19 of the Revised Manuscript).

Reviewer #3

This work developed two novel propenone-linked COFs based on nucleophilic addition

reaction of ketone-activated α -H with aromatic aldehydes, which is an original strategy for synthesis of sp^2 - c conjugated COFs. Interestingly, the authors applied them in the field of photocatalysis and exhibited intriguing properties, especially PyN-DAB achieved ultra-high removal efficiency for multiple actual uranium-containing water samples, which was significant to the actual environmental protection. Overall, the design of the work was elegant and the data have been clearly presented. Therefore, I think that this work deserves publication in Nature Communication after a minor revision, and some suggestions are listed as follows:

1. The focus of this work is to develop a new strategy for synthesizing sp^2 - c conjugated COFs, but the performance of Py-DAB and PyN-DAB in uranium capture, especially the mechanism of photocatalytic reduction of uranium was extensively discussed. Please make corresponding modifications to highlight the focus.

Response: Thanks for the reviewer's valuable suggestion. We have made corresponding modifications to highlight the focus of our work, especially the mechanism explanation section. (Please see pages 14-18 of the Revised Manuscript)

2. The results of bandgaps measured by optical properties using the Kubelka-Monk formula in Figure 3b are very different compared with the bandgaps calculated by DFT in Figure 4, Please provide a reasonable explanation. In addition, Can Figure 4 be merged with Figure 3? Or put Figure 4 in supporting information?

Response: Thanks for the reviewer's valuable suggestion. Generally, COFs are a planar extended π -conjugated skeleton structure, but we only selected a repetitive unit part within the whole framework (i.e. the condensation products of two molecules) for calculation, which is the reason for the difference. However, based on calculations by DFT, the products after Claisen-Schmidt condensation have a narrower band gap compared to the monomers. According to the reviewer's suggestion, we have adjusted Fig. 3 and put Fig. 4 in Supporting Information. (Please see Fig. 3 and Fig. S22).

3. As is well known, vanadium often causes strong interference in the uranium removal process. However, as shown in Figure 5d, vanadium hardly causes interference in the uranium removal process by PyN-DAB. Please explain reason.

Response: Thanks for the reviewer's valuable suggestions. Amidoxime-based adsorbents have become a popular material for uranium adsorption in recent years. However, the

cyclic imide-dioxime form generated during the amide oximation process has a strong binding ability on vanadium. In addition, the broad pH region where the V(V) complex is stable also helps to explain the difficulty of removing vanadium from amidoxime-based sorbents during the elution process (*Nat. Commun.* 2017, 8, 1560), which seriously affects the performance of this type of adsorbent. Currently, sp^2 -c linked COFs adsorbents of uranium were mainly synthesized through Knoevenagel reaction, and then the cyano group was amidoximized through post modification (*Angew. Chem. Int. Ed.* 2020, 59, 17684; *Nat. Commun.* 2020, 11, 436). This process will lead unavoidable vanadium interference. In our work, highly selective synergistic complexation of uranium was achieved through adsorption sites constructed by nitrogen and oxygen atoms in the skeleton (*Angew. Chem. Int. Ed.* 2020, 59, 4168; *Chem. Commun.* 2020, 56, 880-883; *Appl. Catal. B-Environment*, 2021, 294, 120250), thus avoiding the use of amide oxime groups and thus avoiding interference caused by V. At the same time, this strategy allows for the construction of a large number of adsorption sites in one step reaction, which can avoid problems such as decreased crystallinity and BET caused by the complex post modification of cyano COFs through Knoevenagel reaction. Discussions have been supplemented in the Revised Manuscript. “Among them, amidoxime-based COFs are widely used for uranium extraction due to their abundant binding sites, regular pore structure and large specific surface area,^[37] but the process of amidoxime always leads unavoidable vanadium interference.^[21] The uniformly distributed C=O and pyridine nitrogen groups can specifically complex with uranium, which can overcome the shortcomings of amidoxime-based adsorbents.^{[22,25]” (Please see page 14 of the Revised Manuscript).}

4. To prove the introduction of ketone structure can improve the electronegativity of the COFs, the authors selected a repeating unit for DFT theoretical calculation to indicate their surface electrostatic distribution. Zeta potential should also be added.

Response: Thanks for the reviewer’s valuable suggestion. Zeta potential of Py DAB and PyN DAB has been tested to prove the actual electronegativity of the material. The results proved that both Py-DAB and PyN-DAB have more negative potential compared with the amine-linked COF (Py-PD), indicating that the introduction of propenone structure can increase the electronegativity of materials skeleton. Relevant discussions have been supplemented in the Revised Manuscript. “In addition, the test results of the Zeta

potentials proved that both Py-DAB and PyN-DAB have more negative potential compared to the amine-linked Py-PD, indicating that the introduction of propenone structure can increase the electronegativity of materials skeleton (Fig. S29), which is beneficial for promoting the adsorption of UO_2^{2+} .” (Please see page 15 of the Revised Manuscript and Fig. S29 in the Supporting Information).

5. Some graphic details need to be checked and corrected, such as the deformation of “N₂ uptake cm³ g⁻¹” in Figure 2d, and the horizontal axis in Figure 5a. Meanwhile, the type represented by the color of the atom should be indicated in Figure 1a and 1b.

Response: We have carefully checked and corrected the graphic details of the manuscript and supplemented more details of the captions (Please see Fig. 3d, Fig. 5a, captions of Fig. 2a and Fig. 2b in the Revised Manuscript).

REVIEWERS' COMMENTS

Reviewer #1 (Remarks to the Author):

The authors revised the manuscript carefully. I would like to recommend for publication.

Reviewer #2 (Remarks to the Author):

The authors addressed mostly of my concerns properly, but I am still a bit doubt on the response of question 19 that narrowing of the bandgap can directly facilitate the electron transfer, a detailed analysis on the driving force for the charge transfer should also be presented. In addition, there seems no figures showing up for SI1, 2, 3, please fix it. After all the above issues handled, this paper can be published.

Reviewer #3 (Remarks to the Author):

The authors have clarified all my concerns and addressed the questions properly in the revised version. Thus, I think it is now suitable for acceptance.

Response to Reviewers (NCOMMS-23-10968A)

The authors are very grateful to the reviewers for their comments. We have studied comments carefully and responded to the comments point-by-point. The responds to the reviewer's comments are as following.

REVIEWERS' COMMENTS

Reviewer #1 (Remarks to the Author):

The authors revised the manuscript carefully. I would like to recommend for publication.

Reply: Thank you.

Reviewer #2 (Remarks to the Author):

The authors addressed mostly of my concerns properly, but I am still a bit doubt on the response of question 19 that narrowing of the bandgap can directly facilitate the electron transfer, a detailed analysis on the driving force for the charge transfer should also be presented. In addition, there seems no figures showing up for S11, 2, 3, please fix it. After all the above issues handled, this paper can be published.

Reply: Thanks for the reviewer's valuable suggestions. According to the reviewer's comments, we have carefully checked and revised the descriptions of the theoretical calculation section in the Revised Manuscript. "... The results show the bandgap narrowing with the formation of Py-DAB and PyN-DAB (Supplementary Fig. 23). Meanwhile, the holes and electrons in Py-DAB and PyN-DAB are significantly separated (Supplementary Figs. 24, 25), which promotes the occurrence of electron transfer at the interface between the donor and acceptor. ...” (Please see page 13 in the Revised Manuscript and Supplementary Figs. 23-25). In addition, we have corrected the format of the original figures S1, S2 and S3, and now they are correctly displayed (Please see Supplementary Figs. 2, 3 and 4 in the revised Supplementary Information).

Reviewer #3 (Remarks to the Author):

The authors have clarified all my concerns and addressed the questions properly in the revised version. Thus, I think it is now suitable for acceptance.

Reply: Thank you.